# Versatile kit of robust nanoshapes self-assembling from RNA and DNA modules

Alba Monferrer[1], Douglas Zhang[1], Alexander J. Lushnikov[2] & Thomas Hermann [1,3]

DNA and RNA have emerged as a material for nanotechnology applications that take advantage of the nucleic acids' ability to encode folding and programmable self-assembly through mainly base pairing. The two types of nucleic acid have rarely been used in combination to enhance structural diversity or for partitioning of functional and architectural roles. Here, we report a design and screening strategy to integrate combinations of RNA motifs as architectural joints and DNA building blocks as functional modules for programmable self-assembly of a versatile toolkit of polygonal nucleic acid nanoshapes. Clean incorporation of diverse DNA modules with various topologies attest to the extraordinary robustness of the RNA-DNA hybrid framework. The design and screening strategy enables systematic development of RNA-DNA hybrid nanoshapes as programmable platforms for applications in molecular recognition, sensor and catalyst development as well as protein interaction studies.

[1] Department of Chemistry and Biochemistry, University of California, San Diego, 9500 Gilman Drive, La Jolla, CA 92093, USA. [2] University of Nebraska Medical Center, Omaha, NE 68198, USA. [3] Center for Drug Discovery Innovation, University of California, San Diego, 9500 Gilman Drive, La Jolla, CA 92093, USA. Correspondence and requests for materials should be addressed to T.H. (email: tch@ucsd.edu)

Nucleic acid nanotechnology aims to design and build functional materials and devices that self-assemble through base pairing and folding of DNA or RNA strands[1–3]. The fundamental building blocks of nucleic acid architecture, helices and junctions, serve as edges and nodes in the assembly of nucleic acid architectures[4,5] which originate from two fundamentally distinct approaches: the top-down folding of long strands (origami)[6–10], and the bottom-up construction by assembly of autonomously folding RNA modules (Lego[TM])[11,12]. Rich biological functionality of RNA along with the ability to adopt diverse folds within compact motifs have spurred a new field of RNA nanotechnology[13] as well as attempts to create RNA-DNA hybrid nanostructures[14]. While DNA nanomaterials have been explored extensively as devices for protein binding[15], the design of RNA nano-architectures as synthetic scaffolds for protein complex assembly has been approached more recently[16,17]. Past efforts to exploit synergies between RNA and DNA in the design of nanomaterials have largely focused on using extensive strand hybridization between the two types of nucleic acids to create structures that are dominated by RNA-DNA hybrid helices[14].

Here we report a combined design and screening strategy that led to combinations of RNA motifs as architectural joints and DNA building blocks as functional modules for the programmable self-assembly of robust polygonal nucleic acid nanoshapes. The polygonal nanoshapes accommodate topologically diverse DNA modules allowing for broad flexibility to chemically modify, conjugate, or insert protein binding sites, thus furnishing a

versatile kit of nano-scaffolds for applications in molecular recognition, sensor and catalyst development as well as protein interaction studies. The partitioning of architectural and functional roles for RNA and DNA modules in the hybrid nanoshapes provides a general blueprint for expanding chemical diversity and functionality of self-assembling nucleic acid nanomaterials.

## Results

**Design and screening for self-assembling RNA-DNA nanoshapes.** The design strategy for RNA-DNA hybrid nanostructures pursued here seeks to connect RNA motifs that adopt rigid, structurally well-defined folds as topology-defining joints with diverse DNA modules that provide stable components for chemical modification and inclusion of protein binding sites. As key attributes of the RNA-DNA hybrid nanostructures, we aimed for a robust one-step self-assembly process without the need for extensive annealing, high yield efficiency for conversion of nucleic acid modules to assemblies at moderate temperatures, and a set of simple design parameters for facile modification of modules. Our goal was to establish an open platform nano-architecture of self-assembling nucleic acid nanostructures for application and modification by a wide range of users who are not required to be experts in nucleic acid structure or nanotechnology.

We had previously used crystal structure-guided modeling to design triangle and square nanostructures which self-assembled from bent double-stranded RNA motifs carrying complementary overhang sequences of 4 nucleotides only (Fig. 1a, b)[18,19]. While

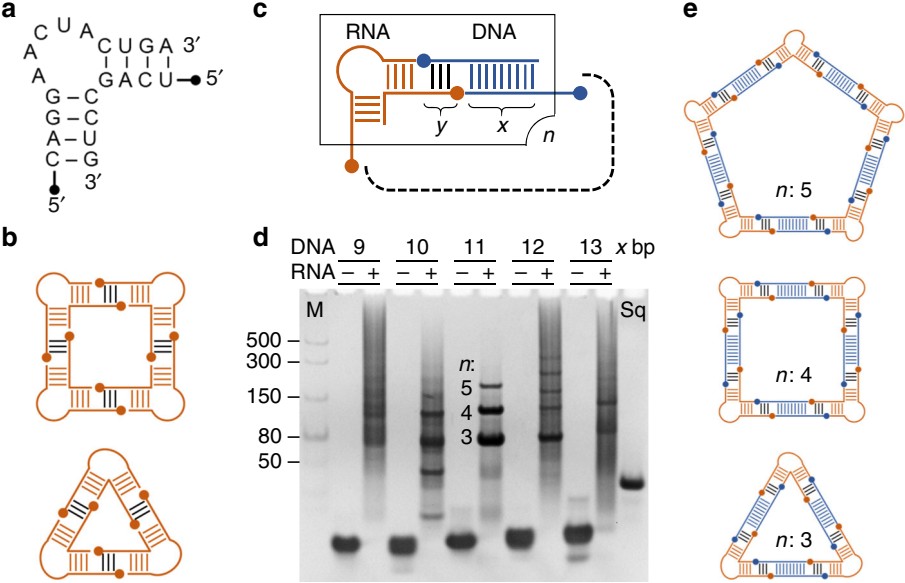

**Fig. 1** Design and screening strategy for RNA-DNA hybrid nanoshapes. **a** Secondary structure of an internal loop RNA from the genome of the hepatitis C virus which adopts a bent fold similar to motifs previously used for the design of all-RNA nanoshapes[23]. **b** Self-assembling all-RNA nanoshapes which were designed from crystal structures of bent internal loop motifs[18,19]. **c** Design of polygonal RNA-DNA hybrid nanoshapes that contain bent RNA motifs as architectural joints and straight DNA modules as connectors. Length variation of single-stranded regions (*y*) for interaction between the building blocks, length of the double-stranded core (*x*) of the DNA modules, and choice of a bent RNA motif give rise to a complex screening library of RNA and DNA components. **d** Screening for stable hybrid complexes of RNA and DNA module combinations by native polyacrylamide gel electrophoresis (PAGE). Stable polygonal nanostructures that self-assemble from a given combination of RNA and DNA modules are expected to give rise to a small number of discrete bands with negligible background and without nucleic acid material retained in the gel pocket. Here, the internal loop RNA shown in **a** with single-stranded overhangs of 6 nucleotides (*y*) was combined with different DNA modules carrying 9–13 base pairs (bp) in the double-stranded core (*x*). For each combination, the DNA module alone (left lane) and together with the RNA motif (right lane) is shown. M is a linear double-stranded RNA size marker. Sq is an all-RNA square comprised of 100 nucleotides which self-assembles from four internal loop RNA modules similar to the one shown in **a** and for which a crystal structure has been determined previously. The three distinct bands observed for the combination of the bent RNA motif with a DNA module containing a core of 11 bp indicate the formation of polygonal nanoshapes (RNA-DNA-11) whose size is consistent with triangles, squares and pentagons. **e** Composition of the RNA-DNA-11 hybrid nanoshapes discovered by screening, including triangle (201 nucleotides), square (268 nucleotides) and pentagon (335 nucleotides). Source data are provided as a Source Data file

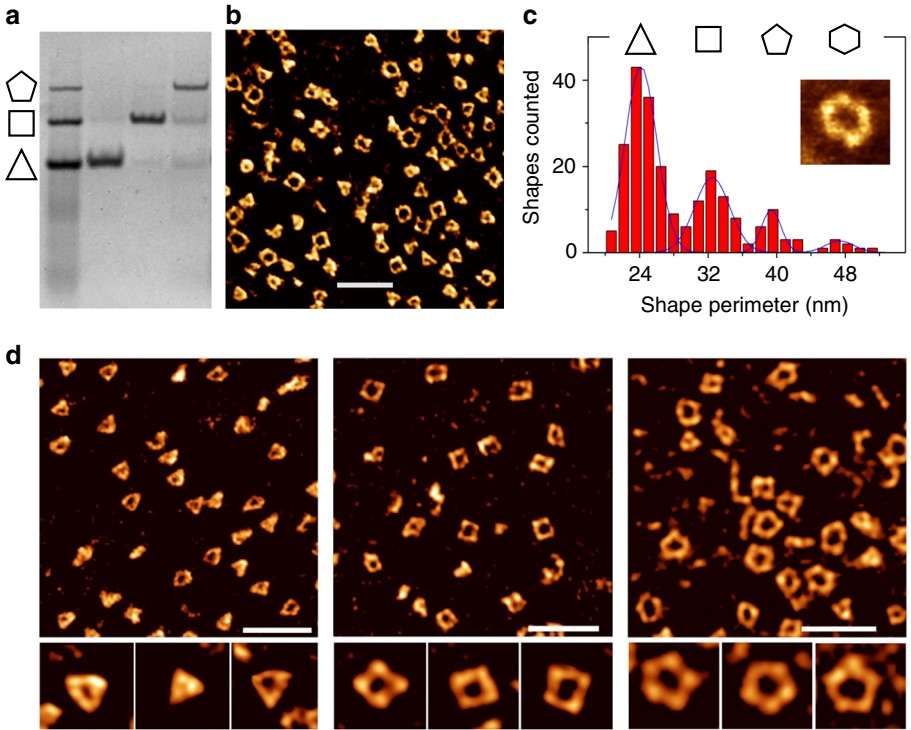

**Fig. 2** Isolation and imaging of RNA-DNA hybrid nanoshapes. **a** Native PAGE analysis of polygonal RNA-DNA-11 hybrid nanoshapes obtained from screening (see Fig. 1d). Nanoshapes isolated by extraction of individual bands from a gel migrate as discrete stable species. **b** Atomic force microscopy (AFM) imaging of the nanoshape mixture. Scale bar represents 50 nm. **c** Perimeter analysis of polygonal nanoshapes observed with AFM imaging. Histogram peaks at multiples of 8 nm indicate the formation of triangles, squares, pentagons and hexagons having a side length that is consistent with the estimated length which includes RNA corner motifs (Fig. 1a) connected through 6 nucleotide single strand hybridization with a linear DNA module of 11 bp. The frequency of observed nanoshapes corresponds to the band intensity of species on the gel (see Fig. 1d). Hexagons occur at a low concentration and are not observed on the gel but are confirmed by AFM imaging (insert). **d** AFM imaging of homogenous populations of nanoshapes obtained by extraction of individual bands from the gel. Scale bars represent 50 nm. Inserts with individual nanoshapes are 30 nm wide. Source data are provided as a Source Data file

nucleic acid helices involving such short sequences are marginally stable at room temperature[20], formation of overall circularly closed structures contributes stabilization through extensive continuous stacking interactions in the resulting polygons[21]. We hypothesized that, among combinations of rigid RNA corner motifs with linear DNA inserts, stable assemblies will be preferred that allow formation of circularly closed, polygonal RNA-DNA nanostructures for which we adopted the term nanoshapes[22] (Fig. 1c).

Combinations of a fixed RNA corner module with doubled-stranded DNA inserts of increasing length were tested for stable complex formation (Supplementary Figs. 1, 2). The RNA corner building block was derived from a noncoding RNA in the internal ribosome entry site (IRES) of hepatitis C virus (HCV)[23] which we previously used to construct and determine the crystal structure of a self-assembling RNA nanosquare comprised of 100 nucleotides (Fig. 1a, Supplementary Fig. 1)[18]. Both, RNA corner module and DNA inserts carried single-stranded overhang sequences of 6 nucleotides for hybridization between building blocks. Combinations of RNA corner and DNA inserts were screened by native polyacrylamide gel electrophoresis (PAGE). A single combination of the RNA corner module with a DNA insert containing 11 base pairs (bp) (RNA-DNA-11) produced clean discrete bands, which indicated the formation of stable assemblies from the nucleic acid components (Fig. 1d). Shorter or longer DNA inserts, ranging from 6 to 15 bp, produced less stable assemblies. The electrophoretic mobility of the three discrete bands observed for the RNA-DNA-11 combination in comparison to a double-stranded RNA size marker was consistent with the formation of compact triangle, square and pentagon RNA-

DNA hybrid nanoshapes (Fig. 1e). The simple screening approach by PAGE is a general and robust method to rapidly assess combinations of nucleic acid modules for self-assembly resulting in stable nucleic acid nanostructures.

In a mixture of inserts with 11 and 6 bp (DNA-11 and DNA-6), only the DNA-11 was selectively incorporated with the RNA corner module to form the three discrete RNA-DNA-11 assemblies (Supplementary Fig. 3). Formation of the discrete RNA-DNA-11 assemblies was resilient to chelation of divalent cations and tolerated the presence of monovalent salt and polyethylene glycol as well as the substitution of magnesium with cobalt(III) hexammine (Supplementary Fig. 4). Heating of the RNA-DNA-11 combination resulted in a single melting transition at 47 °C, which is well above the dissociation temperature expected for the constituent modules including the 11 bp core of the DNA insert and the 6 bp hybrid regions between modules and suggests that dissociation of the three discrete assemblies occurs in concert with strand melting of the RNA and DNA modules (Supplementary Fig. 5).

**Isolation and characterization of nanoshapes.** The kinetic and thermodynamic stability of the RNA-DNA-11 assemblies was demonstrated by extraction of individual bands from a gel and re-analysis by PAGE showing stable single bands (Fig. 2a). Gel purification provided pure samples of individual RNA-DNA-11 assemblies for imaging by atomic force microscopy (AFM). In agreement with PAGE analysis including RNA size marker, AFM imaging of the RNA-DNA-11 combination revealed a mixture of clean polygonal nanoshapes, ranging from triangles to hexagons,

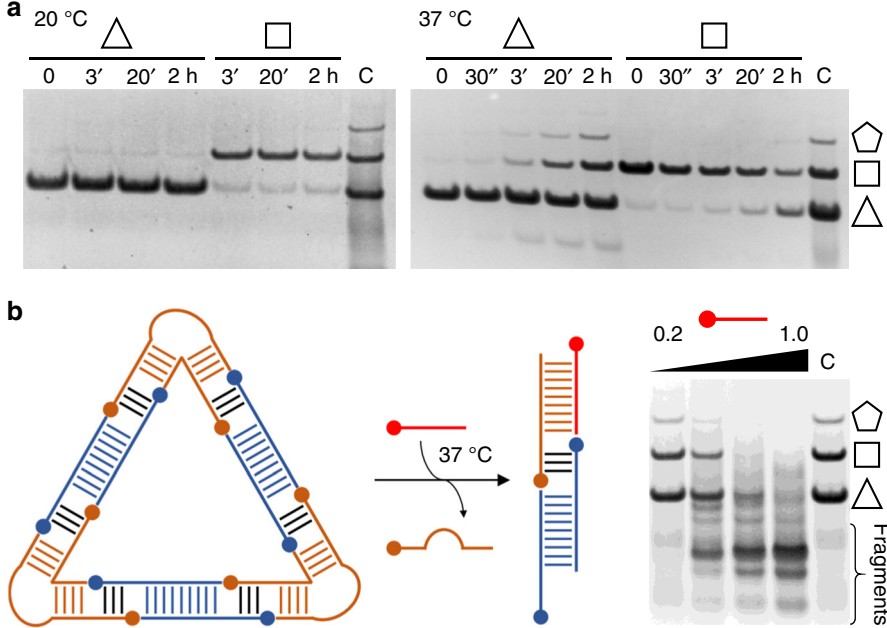

**Fig. 3** Stability and controlled disassembly of RNA-DNA hybrid nanoshapes. **a** Temperature-dependent stability over time of purified RNA-DNA-11 hybrid nanoshapes (see Fig. 2a) analyzed by native PAGE in the presence of 2 mM magnesium salt. Discrete triangles and squares are stable at 20 °C but slowly re-equilibrate to the mixture of nanoshapes at 37 °C. C is a control containing the mixture of nanoshapes before separation of discrete structures (see Fig. 1d). **b** Polygonal nanoshapes, such as the triangle shown as an example, are disrupted by incubation at 37 °C with antisense RNA (red) that is fully complementary to the inner RNA strand. Mixtures of nanoshapes incubated with increasing amounts of antisense RNA were analyzed by PAGE (nanoshape:antisense ratios of 1:0.2, 1:0.5, 1:0.75, 1:1). Lower molecular weight fragments appear concurrently with the disassembly of the nanoshapes. Source data are provided as a Source Data file

devoid of larger aggregates (Fig. 2b, Supplementary Fig. 6). Perimeter analysis of nanoshapes observed by AFM revealed discrete peaks representing polygon classes, from triangles to hexagons (Fig. 2c). The peak spacing at multiples of 8 nm corresponds to the side length of polygons assembled from RNA corner modules that are connected by a DNA-11 insert. Each side of these nanoshapes contains a total of 31 bp, including 8 bp of RNA-RNA double strand combined in the two corner modules, 11 bp of DNA-DNA in the DNA-11 insert and 12 bp of RNA-DNA hybrid combined in the two overlapping regions between insert and corners (Supplementary Figs. 1, 2). The polygon side length of 8 nm as measured by AFM suggests that all bp in the nanoshape sides adopt the A form conformation with a characteristic rise/bp of 0.26 nm[24], amounting to a predicted length of 8.1 nm for a helix of 31 bp.

The frequency distribution of polygons counted in the perimeter analysis from AFM imaging (Fig. 2c) corresponds to the band intensity of discrete assemblies observed in the PAGE analysis of RNA-DNA-11 (Fig. 1d). Hexagon nanoshapes, while occurring at a concentration that was too low for observation on the gel, were detected by AFM imaging and perimeter analysis. Imaging of samples obtained by individual extraction of the three discrete bands from the gel analysis of the RNA-DNA-11 assemblies showed homogenous nanoshape populations of triangles, squares and pentagons, respectively (Fig. 2d), and confirmed the initial correlation of bands to nanoshapes based on electrophoretic mobility comparison to RNA size marker (Fig. 1d, e).

**Control of stable assembly and size of nanoshapes**. Incubation of gel-purified RNA-DNA-11 nanoshapes over 2 h showed that the assemblies were completely stable at 20 °C (Fig. 3a, Supplementary Fig. 7). Slow re-equilibration of pure polygon species to the mixture of nanoshapes was observed above 30 °C, reaching completion within minutes at 37 °C. The identical distribution of

polygon products observed in both samples of de novo annealed nucleic acid components and samples obtained from equilibration of pure polygons suggests that the observed frequency of species may reflect the population of conformational states for the RNA corner motif. We previously demonstrated that the bent fold of the RNA corner module (Fig. 1a) is conformationally flexible and functions as a ligand-captured switch involved in translation initiation by the HCV IRES[25,26]. Extended conformations of the RNA corner motif can be captured in solution by selective small molecule ligands that act as inhibitors of the HCV IRES[27]. The inhibitory action of the ligands is rooted in their ability to increase the population of extended conformations of the RNA corner motif which eventually stalls IRES-driven translation initiation[28]. Therefore, we tested if the population of nanoshapes could be shifted to larger polygons by adding a selective ligand of the RNA corner motif during annealing of the RNA-DNA-11 assemblies. As was expected, hexagon and heptagon nanoshapes appeared in concert with a decreased occurrence of the smaller polygons when RNA-DNA-11 nanoshapes were annealed in the presence of a selective 2-amino-benzimidazole ligand (Supplementary Fig. 8). This first successful example of modulating the formation of nucleic acid nanostructures by a small molecule ligand will provide guidance for the design of ligand-responsive nanoshapes that include aptamer-like motifs.

As an extension of modulating nanoshape formation by a small molecule ligand, we demonstrated that antisense RNA that is complementary to a component strand of the nucleic acid modules could be used to disassemble the RNA-DNA hybrid nanoshapes. Addition of stoichiometric amounts of a 14-nucleotide antisense RNA, which sequesters the inner strand of the RNA corner motif, led to rapid and complete disassembly of the polygonal nanoshapes at 37 °C (Fig. 3b). The ability to control both formation and disassembly of the RNA-DNA hybrid

structures at physiological temperature is an attractive feature for the use of the nanoshapes in biological systems. Such applications may also require to adjust the stability of the nanoshapes. As a proof-of-concept, we modified the RNA-DNA-11 design by extending the length of the single-stranded overhang sequences of the nucleic acid modules from 6 to 7 nucleotides with the intent to increase the stability of the resulting nanoshapes (Supplementary Fig. 9a). To compensate for the increased length of the connecting RNA-DNA hybrid regions, we reduced the number of base pairs in the DNA inserts accordingly. Increased stability of the modified RNA-DNA hybrid nanoshapes was attested by their ability to accommodate a wider range of DNA inserts with 8–10 bp (Supplementary Fig. 9a). Adjustment of the unpaired single strands that connect RNA and DNA modules provides a flexible design parameter for modulating the stability of the nanoshapes that is amenable for variation over a wide range. For example, clean formation of RNA-DNA nanoshapes was observed with designs including further extension of the single-stranded overhangs to 8 nucleotides or shortening to 5 nucleotides, in concert with adjustment of the double-stranded region to 7 or 13 base pairs, respectively (Supplementary Fig. 9b).

Exceptional size scalability of the RNA-DNA hybrid assemblies was demonstrated by designing polygonal nanoshapes with longer sides through the extension of the double-stranded region in DNA inserts (Fig. 4). Multiples of 10 base pairs were added to the original 11 bp DNA module, corresponding to insertion of 1 to 4 turns of the nucleic acid helix. Polygonal RNA-DNA nanoshapes were obtained by self-assembly with inserts of up to 41 bp (4 helix turns), while aggregation occurred with longer DNA modules (Fig. 4a). Progression in scale of the extended-size

nanoshapes with increasing number of helical turns in the DNA inserts was clearly demonstrated by AFM topography which revealed nanotriangles and nanosquares having dimensions in agreement with the expected side lengths (Fig. 4b). The ability to include up to 4 helix turns in the DNA inserts along with variation in length of the single-stranded overhangs for connection to RNA modules enable an exceptionally flexible design approach to fine-tune dimensions and stability of the self-assembling RNA-DNA hybrid nanoshapes.

**Directed assembly of homogenous nanoshapes.** While we found the frequency distribution of polygon species determined by conformational properties of the RNA corner motif, the modular design of the RNA-DNA hybrid nanoshapes allows for modifications that enable controlled assembly of homogenous polygon populations. By introduction of a longer DNA guide strand that contains a defined number of hybridization sites, spaced by a 16 nucleotide linker sequence, homogenous nanoshapes were selectively assembled, including triangles, squares and pentagons (Fig. 5a, b). Both, gel analysis and AFM imaging showed homogenous nanoshapes as the preferred products (Fig. 5b, c). Controlled formation of nanosquares was also achieved through an alternative approach by adding a DNA guide strand with two hybridization sites only (Supplementary Fig. 10a). Rapid incorporation of two DNA guides was preferred, leading to clean formation of nanosquares. Aggregation by potential sharing of the guide DNA between assemblies was not observed for either of the tested variants containing 2–5 hybridization sites, and neither was formation of smaller polygons by incomplete usage of

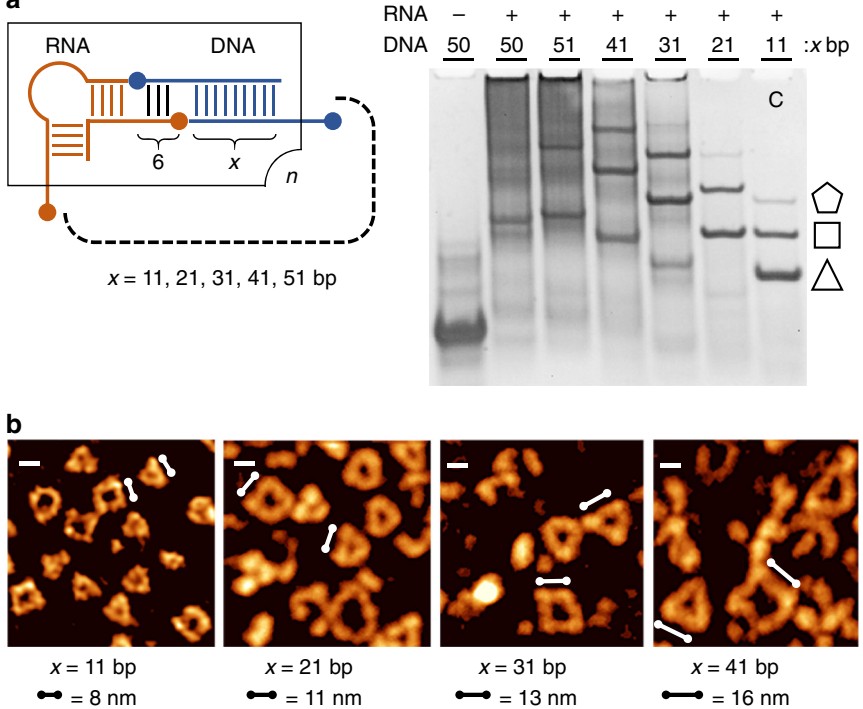

**Fig. 4** RNA-DNA hybrid nanoshapes with extended DNA inserts. **a** The original RNA-DNA-11 nanoshape design obtained from screening (see Fig. 1d) was modified by extending the 11 bp double-stranded region of the DNA inserts by multiples of 10 base pairs corresponding to stepwise addition of 1–4 helix turns. Clean nanoshapes were formed with inserts up to 41 bp while an insert with 51 bp showed instability and aggregation in native PAGE analysis. C is a control containing the originally obtained mixture of nanoshapes that contain a DNA module with 11 bp in the double-stranded region (see Fig. 1d) A 50 bp DNA insert without RNA corner modules is shown for size comparison. **b** AFM topography images of hybrid nanoshapes with DNA inserts from samples used in PAGE analysis in panel a show the size increase of polygons in accord with extension of the base paired region of the DNA modules. Scale bars in upper left corner represent 10 nm. Scale bars with circular ends next to nanotriangles and nanosquares are shown corresponding to the approximate side length expected for nanoshapes containing the extended DNA inserts in all A-form conformation. Source data are provided as a Source Data file

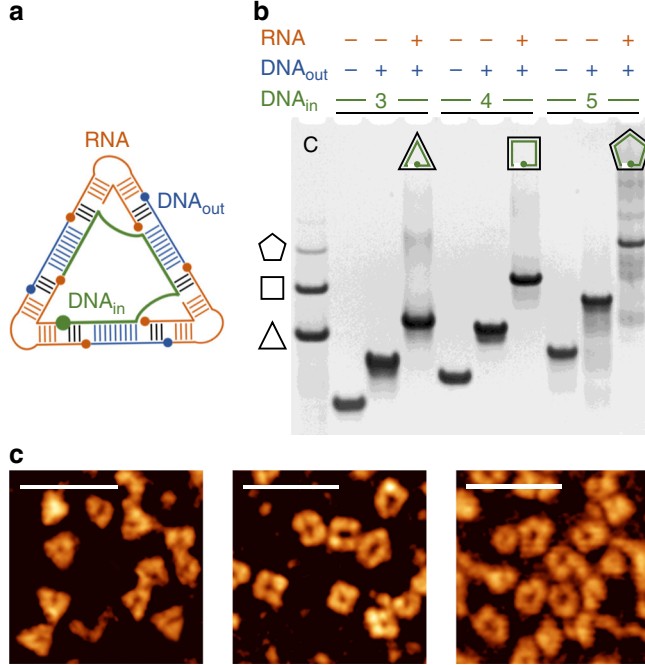

**Fig. 5** DNA modules for controlled assembly of RNA-DNA hybrid nanoshapes. **a** Inclusion of a long DNA guide strand (green) that contains a defined number of hybridization sites, spaced by a linker sequence, directs the formation of homogenous polygonal nanoshapes, as shown here for the triangle. **b** Native PAGE analysis reveals homogenous nanoshapes migrating slightly slower than the same species in the nanoshape mixture shown as control in lane C. **c** AFM topography images of homogenous nanoshapes. Scale bars represent 50 nm. Source data are provided as a Source Data file

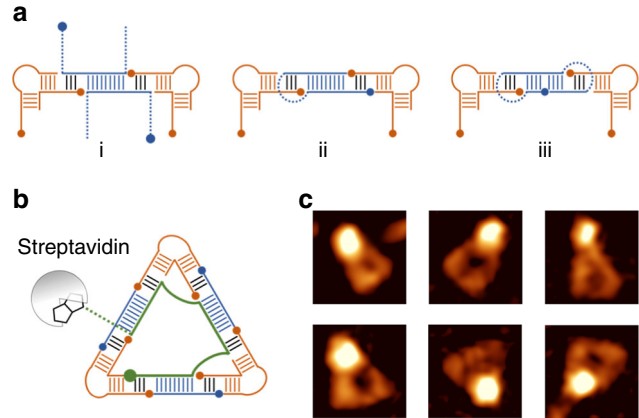

**Fig. 6** DNA modules for functionalization of RNA-DNA hybrid nanoshapes. **a** Topologically diverse DNA building blocks for functionalized hybrid nanoshapes include single strand overhangs on either or both 5′ and 3′ termini (i), monopartite hairpin DNA (ii) and their circular permutation (iii). **b** Design of a homogenous triangle with a DNA guide strand (green) that contains three hybridization sites along with a 3′ single strand extension conjugated with biotin for streptavidin binding. **c** AFM topography images of homogenous triangles with bound streptavidin. Image width is 30 nm

hybridization sites in the guide strand. Subsequent length exploration of the sequences connecting hybridization sites in guide DNAs revealed that linkers as short as 6 nucleotides permitted formation of clean nanoshapes while a shorter connection by 4 nucleotides was not tolerated (Supplementary Fig. 10b).

The homogenous nanoshapes containing a DNA guide strand are stable at room temperature. At 37 °C, where the mixture of RNA-DNA-11 polygons was in a dynamic equilibration, homogenous nanoshapes such as the triangle and square were converted to the mixture of polygons by a short strand carrying a single hybridization site of the DNA-11 insert which displaced the long DNA guide (Supplementary Fig. 11). Conversion of the nanoshapes reached completion within 2 h and proceeded through intermediate assemblies in which the long DNA guide may have been partially displaced by the competing short DNA-11 strand. The complete conversion demonstrates that the originally designed RNA-DNA nanoshapes achieve superior thermodynamic stability despite a potential entropic penalty for complex formation including a larger number of nucleic acid strands than in the homogenous polygons formed with a long guide DNA.

**Diverse and functionalized DNA modules in nanoshapes**. The robust incorporation of a single DNA guide in the homogenous nanoshapes suggested that the RNA-DNA hybrid design might accommodate topologically diverse DNA modules to replace the originally used simple double strands. Incorporation of structurally more complex DNA modules in the nanoshapes will allow the programmable assembly of functional nucleic acid scaffolds for applications involving protein binding and chemical modification. As a proof of concept, we tested assembling nanoshapes

with DNA building blocks containing overhangs on either or both strand termini, hairpins, and their circular permutations (Fig. 6a, Supplementary Figs. 12–14).

Overhang extensions in either or both of the strands of the DNA inserts were tolerated and gave rise to a mixture of nanoshapes similar to the distribution of polygons in the unmodified RNA-DNA hybrid (Supplementary Fig. 12). Extension sequences of the DNA insert strands may be used to include interaction sites for DNA-binding proteins or conjugation to biotin, dyes and other chemical labels. Connection of the strands within the DNA insert gave rise to a hairpin topology which were used to form stable nanoshapes with a distribution of polygons virtually identical to the unmodified RNA-DNA hybrid. Testing of a variety of hairpin loop sizes revealed that loop sizes from 14 down to 2 nucleotides furnished nanoshape mixtures of polygons while connection of the DNA insert strands by a single nucleotide still resulted in formation of the nanotriangles (Supplementary Fig. 13). Connection of the DNA insert strands with hairpins on both sides containing only 2 loop nucleotides afforded by circular permutation of the sequence termini in either strand resulted in modules that were readily incorporated into RNA-DNA hybrid nanoshapes (Supplementary Fig. 14). Clean incorporation of diverse DNA inserts with various topologies attest to the extraordinary robustness of the RNA-DNA hybrid design leading to multi-component nucleic acid nanoshapes.

Modifications of the DNA inserts were applied to guide strands with multiple hybridization sites which we used to control the assembly of homogenous nanoshapes. As a proof of concept, we designed a homogenous nanotriangle that contained a DNA guide carrying a 3′ extended single strand with a terminally conjugated biotin for streptavidin binding (Fig. 6b). The modified homogenous nanotriangle self-assembled as designed and formed discrete streptavidin complexes. At equimolar stoichiometry of the nucleic acid and protein components, predominant complexes were observed that corresponded to one or two nanotriangles associated with one streptavidin (Supplementary Fig. 15). AFM imaging revealed clean complexes of nanotriangles carrying a streptavidin protein attached at a corner position, in agreement with the expected projection of the biotin-conjugated extension sequence of the DNA guide (Fig. 6b, c, Supplementary Fig. 16). Height analysis by AFM demonstrated that these objects

were true complexes conforming with the design of the biotin-conjugated nanotriangles and not fortuitous aggregates of adjoining protein and nanoshape. The measured height of complexes suggested that streptavidin associated at the corner of nanotriangles was resting on top of the nucleic acid assembly as attested by comparison with the height of unbound streptavidin and free nanotriangles (Supplementary Fig. 17).

The clean formation of uniform streptavidin complexes demonstrates the versatility of the RNA-DNA hybrid design as a robust nanoplatform that tolerates topologically diverse DNA inserts as well as the modification of components. While the proof of concept study relied on biotin-conjugation to create a streptavidin complex, binding sites for both single- and double-strand DNA-binding proteins can be readily engineered through sequence and topology modification of the DNA inserts. Chemical oligonucleotide synthesis allows the introduction of ribonucleotide recognition sites for RNA-binding proteins in the DNA building blocks. The RNA-DNA hybrid nanoshapes provide modular discrete scaffolds with small feature sizes and a defined binding-site stoichiometry unlike previously described self-assembling nucleic acid architectures which either required a protein as an intrinsic part of the fold[17] or resulted in an extended nanostructured material[15,16]. Versatility and robustness of the RNA-DNA hybrid nanoshapes predestines them as synthetic materials for controlled arrangement in the study of proteins that bind to scaffolding DNA or ncRNA for their biological function. Unlike the natural cognate ncRNA scaffolds, the synthetic RNA-DNA hybrid nanoshapes can be engineered to contain a subset of protein binding sites enabling to study the interaction of complex assemblies in a controlled fashion and by combinatorial approaches.

## Discussion

We have developed a versatile kit of self-assembling RND-DNA hybrid nanoshapes as a flexible open platform architecture for nanotechnology applications. The design of the hybrid nanoshapes integrates RNA motifs as architectural joints and DNA building blocks as functional modules for modification by addition of sequences, loops and conjugation sites. The modular blueprint of the RNA-DNA hybrid structures enables extensive control over size, shape, composition and thermal stability of the nanoshapes through design parameters that are amenable to independent optimization. The RNA-DNA hybrid nano-architectures combine benefits arising from the diversity of folds available in autonomous RNA motifs and the straightforward chemical modification of robust DNA building blocks. The partitioning of architectural and functional roles for RNA and DNA modules in the hybrid nanoshapes is a novel approach that provides a prototype for expanding chemical diversity and functionality of self-assembling nucleic acid nanomaterials for applications in molecular recognition, sensor and catalyst development as well as protein interaction studies.

## Methods

**Materials**. HPLC or gel-purified RNA and DNA oligonucleotides were purchased from Integrated DNA Technologies. Stock solutions were prepared by dissolving lyophilized single-stranded oligonucleotides in 10 mM sodium cacodylate buffer, pH 6.5. Streptavidin was purchased from Promega Corporation and used without further purification.

**RNA-DNA hybrid assembly preparation**. RNA-DNA hybrid assemblies were prepared by mixing stoichiometric amounts of oligonucleotide components in 10 mM sodium cacodylate buffer, pH 6.5, containing 2 mM magnesium chloride, followed by annealing at 37 °C for 5 min.

**Gel electrophoresis**. Nucleic acid assemblies were analyzed by polyacrylamide gel electrophoresis (PAGE) on 5% native acrylamide/bisacrylamide (19:1) gels in 2X

MOPS buffer (40 mM 3-morpholinopropane-1-sulfonic acid, 10 mM sodium acetate) containing 2 mM magnesium chloride. Gels (10 × 10 cm) were run for 1.5–2 h at 220 V, 22 mA. Nucleic acid was visualized under UV light after ethidium bromide staining.

**Gel extraction**. Discrete nanoshapes were obtained by extraction of individual bands from PAGE gels. Bands were cut from the gel and divided into smaller pieces. 500 μL of 10 mM sodium cacodylate buffer with 2 mM magnesium chloride were added to extract nucleic acid at 4 °C for 5–7 days. Recovered nucleic acid solutions from band extraction were concentrated on Amicon ultra 0.5 mL centrifugal filters (regenerated cellulose 3000 NMWL).

**Protein binding assays**. Pre-formation of the triangle with a DNA strand that contains three hybridization sites along with a 3′ single strand extension conjugated with biotin was performed at 37 °C for 5 min in 10 mM HEPES buffer, pH 7, with 2 mM magnesium chloride. Streptavidin (0.2 or 1 eq) was added and incubated at 37 °C for 5 min. The conjugated samples were analyzed by 5% native PAGE and AFM.

**Temperature-dependent stability experiments**. Pure samples of the isolated nanoshapes were kept at a given temperature (10, 20, 30, or 37 °C). At desired times (0, 3, 20, 120 min), 3 μL aliquots were removed and placed on ice. Aliquots were analyzed by 5% native PAGE.

**AFM imaging and image analysis**. Freshly cleaved mica was modified with 50 mM solution of 1-(3-aminopropyl)-silatrane (APS) in deionized water by immersing strips for 30 min followed by rinsing with deionized water and drying in an Ar stream[29]. RNA samples were diluted at 4 °C in assembly buffer (10 mM HEPES, pH 7 and 2 mM magnesium chloride) and immediately deposited onto APS-modified mica for 2 min. Typical nucleic acid concentration for deposition was 0.5–1.5 ng/μL. Samples were rinsed briefly with several drops of ice chilled deionized water and dried with a gentle flow of argon. AFM images were collected with a MultiMode AFM Nanoscope IV system (Bruker Instruments) in Tapping Mode at ambient conditions. Silicon probes RTESPA-300 (Bruker Nano Inc.) with a resonance frequency of ~300 kHz and a spring constant of ~40 N/m were used for imaging at scanning rate for about 2.0 Hz. Images were processed using the FemtoScan software package (Advanced Technologies Center). The circumference of the nanostructure was measured and analyzed by tracing the distance along the edge of the shape in the middle of RNA strands.

**Thermal denaturation experiments**. Thermal denaturation melting experiments were performed in a UV-Vis spectrophotometer (Shimadzu UV-2401PC). 900 μL of RNA-DNA hybrid nanoshape sample solution at 0.5 μM concentration was prepared with degassed buffer solution, transferred in a 1 cm quartz cuvette and covered with an oil layer to prevent evaporation. The temperature was raised from 15 to 80 °C at a rate of 0.5 °C per minute using a controlled temperature recirculating water system (Lauda RE 206), measuring the absorption every 1 °C at 260 nm.

**Reporting Summary**. Further information on experimental design is available in the Nature Research Reporting Summary linked to this Article.

## Data availability

All data that support the findings of this study are available with the paper and its Supplementary Information, and from the corresponding author on request. The source data underlying Figs. 1d, 2a, 3ab, 4a and 5b and Supplementary Figs. 3, 4a–c, 7–15 are provided as a Source Data file.

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

## Acknowledgements

We thank A. Krasnoslobodtsev for guidance with AFM imaging and S. Chen for helpful discussions. The work was funded by the National Science Foundation, Chemical Measurement & Imaging Program, grant CHE CMI 1608287 to T.H. A.M. thanks the Fundació Joan Riera i Gubau for fellowship support. D.Z. is grateful for support by the UC San Diego Molecular Biophysics Training Grant through NIH T32 GM008326.

## Author Contributions

A.M. and D.Z. performed research, except AFM experiments which were done by A.J.L. T.H. conceived the research and wrote the manuscript. All authors were involved in data analysis and discussion.

## Additional information

**Competing interests:** The authors declare no competing interests.

