## [Peer Review File · Nature Communications]

Reviewers' comments:

Reviewer #1 (Remarks to the Author):

The authors present a new class of synthetic hybrid RNA-DNA modules for self-assembling discrete 2D shapes with controlled 3-fold, 4-fold, or 5-fold symmetry. The hybrid module consists of multiple RNA corner modules from a noncoding region of the internal ribosome entry site of HCV that are interconnected by DNA duplex edges, offering the ability to also incorporate chemical modifications. They demonstrate using PAGE and AFM that these hybrid RNA-DNA modules self-assemble rapidly into a discrete set of polygons: triangles, squares, and pentagons. They also elucidate the thermodynamic stability of each of these shapes following gel purification, showing that at 37C each individual polygon rapidly re-equilibrates to the full set within ~2 hours, whereas at room temperature they maintain their original shape. The addition of an antisense RNA disrupts the nanoshapes at 37C. Finally, they demonstrate that a 'scaffold-like' guide DNA strand can be used to program specific nanoshapes, and modified edges can be included to functionalize biotin-streptavidin complexes, amongst adding other modifications. The identification of this novel RNA motif that robustly forms discrete shapes with the addition of synthetic DNA strands offers a new class of synthetic nucleic acid nanostructure with useful applications in several areas ranging from sensors to possibly therapeutics. Only minor comments are offered below to try to improve the work.

-- AFM data for the ligand-triggered conformational change structures would be helpful since it's difficult to determine whether the larger molecular weight bands on the PAGE are a form of aggregation or explicitly hexagons and heptagon.

-- On Line 72, the authors write, "The simple screening approach by PAGE is a general, robust and readily scalable method..."; yet PAGE is not generally scalable, but rather laborious and manual, so the authors could clarify or modify this statement.

-- The left part of Figure 3b might be clarified, the meaning of the dashed line is not overly clear, though presumably it refers to hybridization.

-- On Line 77, the authors state that the "formation of the discrete RNA/DNA-11 assemblies was resilient to [...] the substitution of magnesium with cobalt(III) hexamine," but in Supplementary Fig. 4b the gel labeling indicates that addition, not substitution, was performed.

-- The final sentence claims that these shapes can aid "in the study of proteins that bind to scaffolding DNA or ncRNA for their biological function." Yet it's unclear how studying protein binding on such a scaffold is an improvement to studying binding of native free DNA or RNA, so this statement could be clarified.

Reviewer #2 (Remarks to the Author):

This paper presents a design approach for hybrid RNA/DNA nanostructures together with their screening method. The authors use RNA motifs as structural joints that connect DNA duplexes to form polygonal nanoshapes. While the manuscript is interesting, this reviewer concerns about its novelty and advantages compared to the previous methods. Also, physical reasoning about the experimental results is not sufficiently provided. Some of major concerns that need to be properly addressed in the manuscript are as follows.

1. First of all, it is not clear to this reviewer what the main advantages of making nanoshapes using both RNA and DNA are. In fact, polygonal shaping, decoration with hairpin motifs, responsive disassembly, and protein/streptavidin binding have been all achieved using DNA only with higher stability. So, the necessity of introducing RNA motifs is not convincingly demonstrated in the current manuscript. Constructing RNA/DNA hybrids are certainly interesting, and usually pursued to exploit both richer biological functionality of RNA and better programmability and structural stability of DNA. However, here, the authors used RNA motifs as architectural joints and DNA blocks as functional modules. Hence, the potential advantage of RNA/DNA hybrids compared with pure RNA or DNA nanostructures is rather ambiguous.

2. Main shortcoming of the current method is that the length of DNA inserts (or the length/size of assembled polygons) seems highly limited according to Figure 1 and Supplementary Figure 9. This suggests that it would be hard to build a nanostructure with various sizes and/or shapes using the current method, which limits the usefulness of the proposed method compared to the well-established other DNA/RNA architecturing methods. In addition, the analysis for the results is quite lacking. For example, the reason why 11-bp-long DNA inserts showed the highest stability and much less stable structures were obtained using other DNA inserts need to be thoroughly investigated and clearly discussed. Also, it is not provided why the DNA-11 was selectively incorporated in a mixture of inserts with 11 and 6 bp (DNA-11 and DNA-6).

3. I think that one novel point of using RNA corners in this paper might be forming a larger nanoshape using a RNA-binding ligand even though it naturally leads to lower synthesis yields. However, only gel results in Supplementary Figure 8 are available in the current manuscript. It must be checked more rigorously with AFM images for these structures whether they formed properly without defects, aggregations, etc. Otherwise, it's not possible to see what kind of structures was assembled in fact.

4. The authors need to explain why triangles, quadrilaterals, pentagons, etc. mixedly appear. Obtaining multiple structures in one-pot process could be beneficial, but it could be also troublesome if it cannot be controlled precisely. Can you control the ratio of these structures in one solution? Related to it, even if we screen the triangles only (for example) from the solution, the other polygonal shapes would appear again after another thermal cycle (heated to 37 degree and cooled down) limiting its repeated usability in various conditions. More rigorous experimental investigation on its controllability is necessary.

Reviewer #1 (Remarks to the Author):

Rev #1: *"Only minor comments are offered below to try to improve the work."*

Rev #1: *"AFM data for the ligand-triggered conformational change structures would be helpful since it's difficult to determine whether the larger molecular weight bands on the PAGE are a form of aggregation or explicitly hexagons and heptagon."*

Response: AFM data from imaging nanoshapes formed in the presence of ligand were added to Supplementary Figure 8 (panel b and discussion in the legend) and referred to on p.6 of the manuscript text. This response also addresses the point #3 of Reviewer #2.

Rev #1: *"On Line 72, the authors write, "The simple screening approach by PAGE is a general, robust and readily scalable method..."; yet PAGE is not generally scalable, but rather laborious and manual, so the authors could clarify or modify this statement."*

Response: The statement was rewritten to refrain from suggesting that the PAGE method is readily scalable; i manuscript text, p.4: *"The simple screening approach by PAGE is a general and robust method to rapidly assess combinations of ..."*

Rev #1: *"The left part of Figure 3b might be clarified, the meaning of the dashed line is not overly clear, though presumably it refers to hybridization."*

Response: To clarify, Figure 3b was modified by showing specifically a nanotriangle as a representative example of the nanoshapes and a statement included in the legend to state this fact.

Rev #1: *"On Line 77, the authors state that the "formation of the discrete RNA/DNA-11 assemblies was resilient to [...] the substitution of magnesium with cobalt(III) hexamine," but in Supplementary Fig. 4b the gel labeling indicates that addition, not substitution, was performed."*

Response: A new panel was added to Supplementary Figure 4 that shows experimental data (native PAGE gel) for the substitution of magnesium with cobalt(III)hexammine (Supp. Fig. 4c). We apologize for the oversight of not including this gel before.

Rev #1: *"The final sentence claims that these shapes can aid "in the study of proteins that bind to scaffolding DNA or ncRNA for their biological function." Yet it's unclear how studying protein binding on such a scaffold is an improvement to studying binding of native free DNA or RNA, so this statement could be clarified."*

Response: To clarify our statement on the use of the hybrid nanoshapes as scaffolds for protein binding, we added a sentence on p.11 of the manuscript: *"Unlike the natural cognate ncRNA scaffolds, the synthetic*

RNA/DNA hybrid nanoshapes can be engineered to contain a subset of protein binding sites enabling to study the interaction of complex assemblies in a controlled fashion and by combinatorial approaches.”

Reviewer #2 (Remarks to the Author):

Rev #2: *“Some of major concerns that need to be properly addressed in the manuscript are as follows.”*

Rev #2: *“First of all, it is not clear to this reviewer what the main advantages of making nanoshapes using both RNA and DNA are. In fact, polygonal shaping, decoration with hairpin motifs, responsive disassembly, and protein/streptavidin binding have been all achieved using DNA only with higher stability. So, the necessity of introducing RNA motifs is not convincingly demonstrated in the current manuscript. Constructing RNA/DNA hybrids are certainly interesting, and usually pursued to exploit both richer biological functionality of RNA and better programmability and structural stability of DNA. However, here, the authors used RNA motifs as architectural joints and DNA blocks as functional modules. Hence, the potential advantage of RNA/DNA hybrids compared with pure RNA or DNA nanostructures is rather ambiguous.”*

Response: As we outlined in the manuscript, the partitioning of architectural and functional roles for RNA and DNA modules in hybrid nanoshapes allows for a modular design that includes the best of the two nucleic acid worlds: structural complexity and a multitude of rigid autonomously folding motifs in RNA, and the stability and ability to readily chemically modify DNA. The hybrid design including small rigid RNA motifs also allows for generally smaller feature sizes than those achievable with DNA only, as we outline in the last section of the Results. To emphasize these points, we have included pertinent statements in the newly added Abstract and Discussion.

Rev #2: *“Main shortcoming of the current method is that the length of DNA inserts (or the length/size of assembled polygons) seems highly limited according to Figure 1 and Supplementary Figure 9. This suggests that it would be hard to build a nanostructure with various sizes and/or shapes using the current method, which limits the usefulness of the proposed method compared to the well-established other DNA/RNA architecturing methods. In addition, the analysis for the results is quite lacking. For example, the reason why 11-bp-long DNA inserts showed the highest stability and much less stable structures were obtained using other DNA inserts need to be thoroughly investigated and clearly discussed. Also, it is not provided why the DNA-11 was selectively incorporated in a mixture of inserts with 11 and 6 bp (DNA-11 and DNA-6).”*

Response: We added extensive new data that demonstrates the exceptional flexibility in the design approach that allows to fine-tune size, shape and stability of the self-assembling RNA/DNA hybrid nanoshapes.

Specifically:

- 1) We included new data on the size scalability of the RNA/DNA hybrid assemblies by designing polygonal nanoshapes with increasingly longer sides through the extension of the double-stranded region in DNA inserts. These scaled nanoshapes were characterized by both native PAGE and AFM topography imaging, shown in the new Figure 4 and discussed on pp.7-8 of the manuscript.
- 2) We included new data on the adjustment of the unpaired single strands that connect RNA and DNA modules which provides a flexible design parameter for modulating the stability of the nanoshapes that is amenable for variation over a wide range. These results on two new nanoshape designs (“5-13” and “8-7”) are shown in the new Supplementary Figure 9b and discussed on p.7 of the manuscript.
- 3) The reason why certain lengths of DNA inserts (for example, 11bp in our main design) show the highest stability is that in combination with the chosen type of “RNA corner” module, these particular sized DNA modules form a circularly closed shape with the least strain. Because of the different conformational

preferences of double-stranded RNA and DNA, as well as hybrid regions of the two types of nucleic acid, a precise prediction of module sizes that result in circularly closed structures is difficult. Therefore, we used a design and screening approach initially to identify the most stable combinations by native PAGE. This procedure and reasoning are outlined in the Introduction and Results of our manuscript.

Rev #2: *"I think that one novel point of using RNA corners in this paper might be forming a larger nanoshape using a RNA-binding ligand even though it naturally leads to lower synthesis yields. However, only gel results in Supplementary Figure 8 are available in the current manuscript. It must be checked more rigorously with AFM images for these structures whether they formed properly without defects, aggregations, etc. Otherwise, it's not possible to see what kind of structures was assembled in fact."*

Response: We have added the AFM data. See our response to the first point of Reviewer #1, above.

Rev #2: *"The authors need to explain why triangles, quadrilaterals, pentagons, etc. mixedly appear. Obtaining multiple structures in one-pot process could be beneficial, but it could be also troublesome if it cannot be controlled precisely. Can you control the ratio of these structures in one solution? Related to it, even if we screen the triangles only (for example) from the solution, the other polygonal shapes would appear again after another thermal cycle (heated to 37 degree and cooled down) limiting its repeated usability in various conditions. More rigorous experimental investigation on its controllability is necessary."*

Response: As we discuss in the manuscript, a mixture of the nanoshapes is obtained because of the conformational flexibility of the "RNA corner" module (p.6): *"The identical distribution of polygon products observed in both samples of de novo annealed nucleic acid components and samples obtained from equilibration of pure polygons suggests that the observed frequency of species may reflect the population of conformational states for the RNA corner motif. We previously demonstrated that the bent fold of the RNA corner module (Fig. 1a) is conformationally flexible and functions as a ligand-captured switch involved in translation initiation by the HCV IRES"*

- 1) To control the shape of the nanostructures and to obtain homogenous populations, we introduced the DNA guide strand concept which is outlined on p.8-9 of the manuscript and in Figure 5. We indeed demonstrate that purified nanoshapes may be re-equilibrated to a mixture of shapes by heat incubation (Fig. 3a and Supp. Fig. 7). We also show that the stability of the nanoshapes can be increased by extending the single-stranded overlap region between RNA and DNA modules (Supp. Fig. 9).
- 2) In the revised manuscript, we further expanded the scope of the approach to prepare homogenous populations of nanoshapes, firstly, by demonstrating that guide strand DNA with two hybridization sites directs formation of exclusively nanosquares (Supp. Fig. 10a) and 2) and, secondly, by length exploration of the sequences connecting hybridization sites in guide DNAs which revealed that linkers as short as 6 nucleotides permitted formation of clean nanoshapes (Supp. Fig. 10b). These new findings are described in the manuscript text on pp.8-9.

Additional revisions:

- 1) We added an Abstract and Discussion to clarify some of the questions the reviewers raised and to conform with the style requirements of *Nature Communications*.
- 2) We added subheadings to the Results section.
- 3) The previous Figure 4 was separated into the new Figure 5 and Figure 6. Figure 4 is new, as outlined above.
- 4) A new author (Douglas Zhang) has been added to the manuscript. He performed additional experiments to obtain data for the revision of the manuscript.

REVIEWERS' COMMENTS:

Reviewer #2 (Remarks to the Author):

Major comments raised by this reviewer have been properly addressed in the revised manuscript.